# Deregulated Expression of *IL-37* in Patients with Bladder Urothelial Cancer: The Diagnostic Potential of the *IL-37e* Isoform

**DOI:** 10.3390/ijms24119258

**Published:** 2023-05-25

**Authors:** Maria Papasavva, Styliana Amvrosiou, Katerina-Marina Pilala, Konstantinos Soureas, Panayiota Christodoulou, Yuan Ji, Konstantinos Stravodimos, Damo Xu, Andreas Scorilas, Margaritis Avgeris, Maria-Ioanna Christodoulou

**Affiliations:** 1Tumor Immunology and Biomarkers Laboratory, Basic and Translational Cancer Research Center, Department of Life Sciences, European University Cyprus, Nicosia 2404, Cyprus; mp181623@students.euc.ac.cy (M.P.); sa181851@students.euc.ac.cy (S.A.); pa.christodoulou@euc.ac.cy (P.C.); 2Department of Biochemistry and Molecular Biology, Faculty of Biology, National and Kapodistrian University of Athens, 15771 Athens, Greece; pilalakm95@gmail.com (K.-M.P.); kon.soureas@gmail.com (K.S.); ascorilas@biol.uoa.gr (A.S.); mavgeris@med.uoa.gr (M.A.); 3Laboratory of Clinical Biochemistry—Molecular Diagnostics, Second Department of Pediatrics, School of Medicine, National and Kapodistrian University of Athens, “P. & A. Kyriakou” Children’s Hospital, 11527 Athens, Greece; 4School of Medicine, European University Cyprus, Nicosia 2404, Cyprus; 5School of Infection and Immunity, University of Glasgow, Glasgow G12 8TA, UK; y.ji.1@research.gla.ac.uk; 6First Department of Urology, “Laiko” General Hospital, School of Medicine, National and Kapodistrian University of Athens, 11527 Athens, Greece; kgstravod@yahoo.com; 7State Key Laboratory of Respiratory Disease for Allergy Shenzhen University, Shenzhen Key Laboratory of Allergy and Immunology, School of Medicine, Shenzhen University, Shenzhen 518055, China; xdm@szu.edu.cn

**Keywords:** interleukin (IL-)37, bladder cancer, biomarker, survival, advanced stage

## Abstract

Cellular and molecular immune components play a crucial role in the development and perpetuation of human malignancies, shaping anti-tumor responses. A novel immune regulator is interleukin-37 (IL-37), already shown to be involved in the inflammation associated with the pathophysiology of many human disorders, including cancer. The interplay between tumor and immune cells is of great importance, especially for highly immunogenic tumors such as bladder urothelial carcinoma (BLCA). This study aimed to investigate the potential of IL-37 and its receptor SIGIRR (single immunoglobulin IL-1-related receptor) to serve as prognostic and/or diagnostic markers in patients with BLCA. To this end, a series of bioinformatics tools processing -omics datasets and specifically designed qPCR assays on human BLCA tumors and cancer cell lines were utilized. Bioinformatics analysis revealed that *IL-37* levels correlate with BLCA tumor development and are higher in patients with longer overall survival. Furthermore, mutations on *SIGIRR* are associated with enhanced infiltration of the tumor by regulatory T cells and dendritic cells. Based on the qPCR validation experiments, BLCA epithelial cells express the *IL-37c* and *IL-37e* isoforms, while the latter is the predominant variant detected in tumor biopsies, also associated with higher grade and the non-muscle-invasive type. This is the first time, to the best of our knowledge, that *IL-37* and *SIGIRR* levels have been assessed in BLCA tumor lesions, and associations with pathological and survival parameters are described, while a transcript variant-specific signature is indicated to have a diagnostic potential. These data strongly indicate the need for further investigation of the involvement of this cytokine and interconnected molecules in the pathophysiology of the disease and its prospective as a therapeutic target and biomarker for BLCA.

## 1. Introduction

The interleukin (IL)-1 family includes cytokines that can act on innate immune components and alter their survival and function, while they can also regulate adaptive immune responses [1]. One of the most recently identified members with immunosuppressive properties is IL-37. This cytokine is expressed by immune and non-immune cell subsets and functions by suppressing the production of pro- and inducing the activity of anti-inflammatory signals [2]. Similar to the classical immunoregulatory cytokines TGF-β and IL-10, the modulating properties of IL-37 attracted the interest of researchers both from basic and translational points of view [2]. Especially in recent decades, the frontiers of IL-37 exploration are ever-increasing and include work on its involvement in the pathogenesis of human diseases but also its specific expression patterns in certain conditions, including cancer [3].

The *IL-37* gene is located on human chromosome 2q12-13, in close proximity with the regulatory regions of the *IL-1a* and *IL-1b* genes, and codes for five isoforms: a, b, c, d and e [1]. The most well-studied isoform is IL-37b, the longest one consisting of five of the six exons (missing exon 3) [2,3]. Nevertheless, the specific functions and relative abundance of all isoforms have not yet been fully described [4]. It has been reported that IL-37 is constitutively produced by monocytes, macrophages, dendritic cells (DCs), B cells and plasma cells [5,6,7]. Following pro-inflammatory stimuli in tissue cells and peripheral blood mononuclear cells (PBMCs), especially monocytes, the expression of the cytokine can be dramatically enhanced, playing thereafter a fundamental immunoregulatory role in the local microenvironment [2,5,6,8].

IL-37 is known as a dual-function cytokine acting on target cells through two independent mechanisms. The first one takes place extracellularly, where the cytokine binds to IL-18R1 (interleukin 18 receptor 1), which recruits IL-1R8 (interleukin 1 receptor 8), also known as SIGIRR (single Ig IL-1-related receptor). SIGIRR, critical for the anti-inflammatory activity of IL-37, forms a complex with IL-1R8 that transduces the signal intracellularly [8,9,10,11]; mTOR (mammalian target of rapamycin), AKT (Ak strain transforming), STAT1 (signal transducer and activator of transcription 1), SHP-2 (SH2 domain-containing protein tyrosine phosphatase-2), PTEN (phosphatase and tensin homolog), PI_3_K (Phosphoinositide 3-kinases) kinase, MAPK (mitogen-activated protein kinase) and other pro-inflammatory signaling cascades are repressed [4,5]. The second mechanism occurs intracellularly, upon the cleavage of IL-37 by caspase-1 at the aspartic acid (D20). Cleaved IL-37 binds to Smad3 and can be moved into the nucleus, where it dampens the expression of pro-inflammatory genes [12,13,14,15]. Furthermore, extracellular IL-37 is an activator of AMP-dependent protein kinase and an activator of the Mer-PTEN-DOK (docking protein) pathways. In addition, IL-37/IL-18Ra/IL1-R8 retards MyD88 (myeloid differentiation primary response 88) and thereby harnesses the signal transmission downstream TLRs (Toll-like receptor) [16,17]. Within the inflammatory network, IL-37 suppresses the expression of certain pro-inflammatory cytokines, while promoting the expression of anti-inflammatory ones through adjustment of the polarization of macrophages, control of lipid metabolism, impact on the function of inflammasomes and regulation of micro-RNAs (miRNAs) [16].

Based on cumulative data from recent years, IL-37 employs its tumor-regulatory effects to control the functions of other immune and non-immune components of the tumor microenvironment (TME). Here, cellular and non-cellular components (including cytokines, chemokines and growth factors) play a pivotal role in the development, establishment and progression of the tumor, and its response to therapy [17]. Some of these, such as T helper (Th) 1, cytotoxic T, NK, B cells, M1 macrophages (ΜΦ) and mature dendritic cells (DCs), are partners of immune surveillance and act against malignant cells, while others, such as Th2, regulatory T cells (Tregs), M2 ΜΦ, neutrophils, immature DCs, cancer-associated fibroblasts (CAFs) and myeloid-derived suppressor cells (MDSCs), encourage cancer cells’ escape from immune control [18]. Essentially, pivotal components of the TME, including immune checkpoints such as PD-1/PD-L1 (programmed death-(ligand) 1), and angiogenic factors such as VEGF (vascular endothelial growth factor) now serve as routine immunotherapeutic targets associated with good clinical outcomes.

Emerging data from clinical and preclinical studies strongly indicate the important contribution of IL-37 to anti-tumor immunity via control of the function of critical regulators in the TME. In the case of human hepatocellular carcinoma (HCC), TAM (tumor-associated macrophage)-derived IL-37 impedes their polarization towards the M2 phenotype by affecting the IL-6/STAT3 axis, eventually inhibiting the growth of the tumor [19]. The enhancement of IL-37 expression by HCC cells is accompanied by an increase in the levels of CCL3 (chemokine (C-C motif) ligand 3) and CCL20, which further promotes the recruitment of CD1a^+^ dendritic cells (DCs) in the tumor infiltrate [20]. Additionally, recombinant IL-37 hampers the immunosuppressive effect of HCC cells against DCs, which is demonstrated by the increased expression of the surface molecules MHC-II, CD86 and CD40, while it stimulates DCs to secrete IL-2, IL-12, IL-12p70, interferon-a (IFN-α) and IFN-γ, further enhancing the anti-tumor effect of T cells. Furthermore, HCC overexpression of IL-37 is associated with increased recruitment of CD11c^+^ DCs in the tumor infiltration and suppression of the growth of the tumors [20]. However, in an experimental model of colitis-induced colorectal cancer (CRC), IL-37 transgenic mice were more prone to the development of tumors and were also characterized by significantly enlarged tumor burdens and impaired function of CD8^+^ T cells, supporting its tumor promoting effect in this case [21]. Adding to this, the transfection of the *IL-37* gene in Jurkat human CD4^+^ leukemic cells promoted the induction of a Treg phenotype characterized by elevated CTLA-4 and FOXP3 expression and IL-10 secretion, capable of suppressing effector T cell activation and proliferation [22].

During tumor development and growth, IL-37 also acts as an anti-angiogenic factor; expressed by cancer cells, it inhibits the in vitro formation of tubules of human umbilical vein endothelial cells (HUVEC), lowers the expression of matrix metallopeptidase 2 (MMP2) and vascular endothelial growth factor (VEGF) in SK-Hep-1 cells and dampens angiogenesis in the tumors of mice with HCC [23]. Moreover, it can inhibit Rac1-associated migration in various cancer cell types [23,24], while it is also linked to lower metastatic potential in human lung adenocarcinoma (LUAD) [25].

Intriguingly, emerging evidence over the last decade strongly supports the suggestion that IL-37 may serve as a potential biomarker in certain human malignancies. It is interesting that in CRC patients, serum IL-37 levels are increased and associated positively with the serum levels of CEA (carcinoembryonic antigen), a widely-used biomarker for CRC diagnosis and monitoring, and negatively with the infiltration rates of CD8^+^ T cells in the tumor [26]. In these patients, high intratumoral IL-37 expression has been found to characterize early-stage tumors and correlates with good prognosis of disease-free (DFS) and overall survival (OS) [26]. More importantly, along with the incidence of CD66b^+^ neutrophils and mismatch repair (MMR) status, IL-37 levels have been included in nomograms predicting DFS and OS, aiding progression towards more personalized management of CRC patients [26]. Similarly, a high prevalence of IL-37^+^CD1a^+^ DCs in HCC tumor infiltration is a marker of favorable prognosis of patients [20]. We recently showed that high *IL-37* mRNA levels can serve as a favorable factor for the OS of LUAD patients associated with less severe disease, and also with a certain LUAD transcriptomic signature [25].

In contrast, in other malignancies, high IL-37 expression is regarded as an adverse factor since it is associated with poor prognosis and/or more severe disease. Acute myeloid leukemia (AML) individuals exhibit lower levels of serum IL-37 compared to controls, which correlates with unfavorable prognosis, and they are restored in complete remission [27]. Peripheral blood *IL-37* mRNA and CD8^+^ T cell prevalence have been found to be reduced in patients with breast cancer compared to healthy individuals, and are associated with ER^+^/PR^+^/HER2^+^ status [28]. In oral squamous cell carcinoma (OSCC), the ratio of IL-18 to IL-37 levels is increased in the serum and PBMCs of the patients, and is associated with shorter OS and DFS [29]. Lastly, high levels of IL-37 secreted by peripheral blood Tregs of individuals with melanoma reflect the secretion of TGFβ and other IL-1β mediators by the tumor, thus indicating it could be interpreted as a potential marker for immunosuppression induced by the tumor [30].

From all the above, it can be considered that IL-37 may represent a crucial checkpoint adjusting local anti-tumor immune responses, and a potential biomarker for the status and prognosis of disease. Especially for highly immunogenic cancers, the study of this cytokine could reveal useful information underlying their pathogenesis and suggest possible management approaches. Bladder urothelial cancer (BLCA) is one of the most immunogenic human tumors and responds well to immunotherapeutic approaches, such as intravesical bacillus Calmette–Guérin (BCG) and immune-checkpoint inhibitors (ICIs), especially in advanced stages [29,31]. However, as with other tumors, BLCA cells promote the formation of highly immunosuppressive TME, hindering effective anti-tumor immune responses [32]. This constitutes a reasonable background for the exploration of pivotal players within the local immune milieu, such as IL-37, that could be manipulated against tumor growth, similarly to other components of the BLCA TME, e.g., the fibroblast growth factor receptor 3 (FGFR3), that have very recently been suggested as possible therapeutic targets [33]. To date, only one study has been performed on IL-37 in BLCA, suggesting its increased levels in the serum of patients [34]. These limited data intensify the need for further investigation of this cytokine in this particular type of cancer.

Herein, we aimed to investigate the possible prognostic potential of IL-37 in patients with BLCA using a series of bioinformatics tools and publicly available databases, followed by validation using specifically developed qPCR assays. *IL-37* levels were found to be correlated with tumor development, stage, nodal metastasis and mutation status, as well as improved survival of patients. Furthermore, mutations and gene expression levels were associated with the differential distribution of certain immune cells infiltrating the tumor. Validating qPCR experiments revealed that human BLCA biopsies predominantly express the *IL-37e* isoform positively associated with higher tumor grades.

## 2. Results

### 2.1. IL-37 Levels Are Increased in BLCA versus Non-Cancerous Bladder Tissues

The possible differential expression of *IL-37* in cancerous (BLCA) vs. non-cancerous bladder biopsies (non-BLCA) was explored using the UALCAN portal (http://ualcan.path.uab.edu/; accessed on 1 December 2022) [35] and TNMplot web tool (www.tnmplot.com; accessed on 20 December 2022) [36]. IL-37 expression levels were found to be increased 44.43-fold in samples from BLCA (*n* = 411) compared to those from non-BLCA individuals (*n* = 30, Mann–Whitney *p* = 1.44 × 10^−10^), and 35-fold in tumors compared to paired adjacent normal tissues from BLCA individuals (*n* = 19; *p* = 2.26 × 10^−3^) (Figure 1A, Table 1). *SIGIRR* mRNA expression exhibited no change between BLCA and non-BLCA biopsies (fold-change = 1.28; *p* = 3.82 × 10^−1^), while it was slightly increased between tumor and adjacent normal tissues in BLCA patients (fold-change = 1.35; *p* = 1.86 × 10^−2^) (Figure 1A, Table 1).

### 2.2. Increased IL-37 or SIGIRR Expression Are Favorable Prognostic Factors for OS in BLCA Patients

Analysis through the Kaplan–Meier Plotter tool (www.kmplot.com; accessed on 1 December 2022) [37] revealed that BLCA patients who expressed high levels of *IL-37* (*n* = 275) or *SIGIRR* (*n* = 216) have a greater probability for longer OS compared to those with low expression of these genes (*n* = 129 and 188, log rank *p* = 1.8 × 10^−2^ and 1.5 × 10^−5^ for *IL-37* and *SIGIRR*, respectively) (Figure 1B). In detail, high *IL-37* expression increases the probability of surviving up to 150 months by 31% (hazard ratio (HR) = 0.69; 95% CI = 0.51–0.94), and high *SIGIRR* expression by 48% (hazard ratio (HR) = 0.52; 95% CI = 0.39–0.7). The median survival time of patients’ groups was 55.67 months for the high *IL-37*, 23.73 months for the low *IL-37*, 67.33 months for the high *SIGIRR* and 21 months for the low *SIGIRR* cohort.

### 2.3. IL-37 and SIGIRR Levels Associated with Histopathological Parameters of BLCA Tumors

Data processing through the UALCAN portal (http://ualcan.path.uab.edu/; accessed on 20 December 2022) [35] suggested that the levels of *SIGIRR* are slightly higher in BLCA papillary (*n* = 132) compared to non-papillary tumors (*n* = 271) (fold change = 1.25, *p* = 2.78 × 10^−4^) (Figure 2, Table 2). However, *IL-37* levels were similar in the two histological types of BLCA. Importantly, both the levels of *IL-37* and *SIGIRR* were found to correlate with the stage of the BLCA tumor, with lower expression detected in advanced stages (Figure 2, Table 2). Specifically, higher levels of both genes were detected in stage 1 tumors (median (range): for *IL-37*, 1.03 (0–2.06) and for *SIGIRR*, 40.08 (35.86–44.30)); it should be mentioned, however, that only two samples were included in this group. Stage 2 tumors (*n* = 129) exhibited intermediate expression levels (for *IL-37*, 0.1 (0–0.64) and for *SIGIRR*, 33.13 (3.81–84.25)), while stage 3 (*n* = 137) and 4 (*n* = 132) tumors the lowest ones (stage 3: for *IL-37*, 0.08 (0–0.52) and for *SIGIRR*, 26.84 (2.30–70.21), stage 4: for *IL-37*, 0.09 (0–0.73) and for *SIGIRR*, 27.23 (4.33–78.70)). Between cohorts of different stages, noteworthy differences were observed in the cases of *SIGIRR* expression in stage 3 versus stage 2 (fold change = 0.81, *p*= 2.43 × 10^−2^) and stage 4 versus stage 2 (fold change = 0.82, *p*= 2.52 × 10^−2^).

*IL-37* expression levels were also found to correlate with the nodal metastasis status of the tumor, with the highest values detected in N3 biopsies; however, the group included a relatively small number of samples (*n* = 8, 0.26 (0.095)) (Figure 2, Table 2). Finally, levels of both genes were associated with the mutation status of the tumor, yet in an opposite trend: higher levels of *IL-37* were detected in individuals bearing *TP53* (tumor protein 53) mutations (*n* = 193, 0.116 (0–0.779)) and/or *ARID1A* (AT-rich interactive domain-containing protein 1A) mutations (*n* = 76, 0.096 (0–0.824)) compared to those without these mutations (*n* = 215, 0.069 (0–0.616) and 324, 0.091 (0–0.641)), respectively), whereas higher levels of *SIGIRR* were detected in non-mutated (for *TP53*: 29.68 (3.67–71.13) and for *ARID1A*: 29.69 (3.81–82.14)) versus mutated tumors (for *TP53*: 27.84 (2.3–84.45) and for *ARID1A*: 28.93 (3.81–82.13)) (Figure 2, Table 2).

### 2.4. IL-37 and SIGIRR Gene Alterations Correlate with Altered Infiltration of the BLCA by Certain Immune Cell Subset Levels Associated with Histopathological Parameters of BLCA Tumors

The effects of the *IL-37* and *SIGIRR* gene mutations on immune cell infiltration in bladder cancer tumors were evaluated by searching through the “mutation” module of the TIMER2.0 webserver (http://timer.cistrome.org/; data accessed on 20 December 2022) [38]. As revealed, tumors bearing non-synonymous, somatic mutations in the *IL-37* gene were characterized by higher infiltration of B lymphocytes (XCELL project; log_2_fold change = 1.245, Wilcoxon *p* = 0.046), while those bearing non-synonymous, somatic mutations in the *SIGIRR* gene had significantly higher infiltration by Tregs (CIBERSORT project; log_2_fold change = 19.932, Wilcoxon *p* = 0.019), slightly elevated rates of DCs (XCELL project; log_2_fold change = 19.932, Wilcoxon *p* = 0.057) and B cells (CIBERSORT project; log_2_fold change = 1.37, Wilcoxon *p* = 0.029) and decreased incidence of CD4^+^ T cells (XCELL project; log_2_fold change = -2.001, Wilcoxon *p* = 0.037) and endothelial cells (EPIC project; log_2_fold change = −1.585, Wilcoxon *p* = 0.032) (Figure 3A).

### 2.5. IL-37 Expression Levels Correlate with Altered Infiltration of the BLCA by Certain Immune Cell Subsets

As revealed by processing RNA sequencing data through the TIMER2.0 “gene” module, *IL-37* expression levels correlate very weakly with the infiltration rates of CD8^+^ T cells (MCPCOUNTER; Spearman’s rho = −0.146, *p* = 5.14 × 10^−3^), CD4^+^ memory T cells (XCELL; Spearman’s rho = −0.185, *p* = 3.62 × 10^−4^) and neutrophils (TIMER; Spearman’s rho = −0.164, *p* = 1.55 × 10^−3^) in a negative manner, and with those of Tregs in a positive manner (CIBERSORT; Spearman’s rho = 0.14, *p* = 7.00 × 10^−3^) (Figure 3B).

### 2.6. Human Bladder Cancer Cells Express IL-37b, c and e Isoforms

The application of specifically designed qPCR assays indicated that the human T24 and RT4 BLCA cell lines express three of the five IL-37 isoforms, namely *IL-37b*, *c* and *e* (Figure 4A, left). The most prominent expression in both cell lines was that of *IL-37c*, followed by *IL-37e* and finally *IL-37b*. Furthermore, compared to T24 cells, RT4 cells exhibited significantly higher levels of *IL-37c* (8.2 folds; mean ± standard error (SE) for RT4 vs. T24, respectively = 1.23 ± 0.03 vs. 0.15 ± 0.01, *p* = 0.0002]) and *IL-37e* (7.09 folds; 0.78 ± 0.02 vs. 0.11 ± 0.02, *p* < 0.0001). With regard to *IL-37b*, this was higher in T24 than in RT4 cells (7.5 folds; 0.06 ± 0.01 vs. 0.008 ± 0.001, respectively, *p* = 0.034).

### 2.7. Human BLCA Biopsies Predominantly Express the IL-37e Isoform: Correlation with the Grade of the Tumor

Following the same approach in representative specimens, it was observed that *IL-37e* is the predominant transcript variant detected in human tumor biopsies of BLCA patients (Figure 4A, right). *IL-37b* and *c* are marginally expressed, while no expression was detected for the *IL-37a* and *d* variants. Validation in a larger cohort of patients revealed that BLCA biopsies of high grade (*n* = 49) express significantly increased levels of the *IL-37e* variant compared to low-grade ones (*n* = 18) (fold change = 2.79, *p* = 0.0269, mean (range) = 0.39 (0.024–50.83) versus 0.14 (0.007–25.18), respectively) (Figure 4B, Table 2). Indeed, there is an 11.67-fold increase in grade 2 (*n* = 16) over grade 1 (*n* = 6) biopsies (0.35 (0.029–25.18) versus 0.03 (0.007–1.88), *p* = 0.049) and an 11-fold increase in grade 3 (*n* = 45, 0.33 (0.024–50.85)) over grade 1 biopsies (*p* = 0.017). These differences were attributed more to the non-muscle-invasive group, as revealed when this cohort was individually analyzed (Figure 4B, Table 2). Separate analysis of the muscle-invasive group was not applicable due to the lack of specimens of grades 1 and 2.

## 3. Discussion

Bladder cancer (BLCA) accounts for the second most frequent malignancy of the male genitourinary tract, and the sixth most common male malignancy worldwide, associated with high morbidity and mortality rates and costs [39,40]. Environmental or occupational exposures to certain carcinogens, including tobacco, are primary risk factors for the development of BLCA. A suspected diagnosis of the disease usually occurs after macroscopic hematuria as the first clinical symptom and is confirmed upon transurethral resection of bladder tumor (TURBT), which also serves as the first treatment approach [41].

BLCA tumors are classified into non-muscle-invasive papillary (NMIBC; Tis, Ta, T1) and non-papillary muscle-invasive (MIBC; T2-T4) ones, which are associated with distinct pathogenetic mechanisms and have unique pathological features and different molecular characteristics [41,42]. Newly diagnosed NMIBC patients (∼75% of primary BLCA) exhibit frequent relapses (∼50–70%) and progression to muscle-invasive disease (∼15%) [43,44]. On the other hand, primary MIBC (∼25% of BLCA) displays strong metastatic potential and is life-threatening [45]. In metastatic disease, the identification of genetic drivers and advances in immunotherapy are critical weapons for optimized care of patients [44]. Intravesical BCG and the use of the PD-1/PD-L1 inhibitors atezolizumab, durvalumab, avelumab, nivolumab and pembrolizumab are now part of clinical routine practice to treat late-stage patients [45]. Current large trials are assessing the therapeutic potential of novel ICIs and other immunotherapeutic agents, as well as combinations of them with already used drugs for improved clinical outcomes (www.clinicaltrials.gov; data accessed on 1 February 2023).

In spite of the considerable reduction in BLCA-associated mortality of recent years, which is due to developments in diagnostic and therapeutic approaches [45], there is still progress to be made for the advanced manipulation of anti-tumor immune responses. As part of this, this study was focused on the exploration of IL-37 as a possible pivotal partner of the local TME that may serve as a potential biomarker for the disease. By processing a large set of RNA sequencing data through appropriate bioinformatics tools, it was revealed that *IL-37* expression is significantly increased in BLCA vs. non-BLCA tissues, suggesting a disease-specific involvement of this cytokine in the pathological lesion. Within BLCA tissues, those of higher grade are characterized by slightly lower levels of *IL-37* compared to lower-grade ones, as supported by processing publicly available datasets. However, in-house qPCR experiments developed for the assessment of *IL-37e* expression, the predominant transcript variant in BLCA biopsies, support that higher-grade tumors exhibit significantly increased levels of this isoform compared to lower-grade ones. This inconsistency may be due to two factors. First, the TCGA database contains data exclusively from patients with MIBC and none with NMIBC (https://www.cancer.gov/tcga; accessed on 29 December 2022). Our validation cohort included both MIBC and NMIBC (60.34% and 39.66%, respectively) tumors; the first exhibiting 1.6-fold higher levels of *IL-37e* compared to the second. Furthermore, early-stage (stage 1) biopsies are relatively rare within MIBC (in our in silico analysis, 2 out of 400 samples) and, thus, one needs to be cautious when interpreting results for stage 1 data in this cohort. The second factor regards the transcripts analyzed. In the bioinformatics analysis, total *IL-37* levels were evaluated, while the validation qPCR experiments focused solely on *IL-37e*. Given the fact that the in silico results supported marginal changes between early and late BLCA stages, it is quite possible that the differences in *IL-37e* in this cohort are masked.

At this point, it is also worth commenting on the differential expression pattern observed between BLCA tumor biopsies and BLCA cell lines. Our data suggest that BLCA epithelial cells express mainly *IL-37c* and *IL-37e*, with the first in relatively higher levels. However, when the whole tumor (including TME) was analyzed, *IL-37e* was found to be primarily expressed and, indeed, at dramatically greater levels compared to (a) other *IL-37* isoforms and (b) *IL-37e* expressed by BLCA cell lines. This is a strong indication that this mRNA variant is the primary one transcribed in the cells of the TME. However, whether the corresponding protein isoform is produced and actively involved in the local cytokine milieu remains to be clarified. Focusing on bladder cancer epithelial cells, it is also important to compare the expression levels of each of the two mRNA variants between the two cell lines. Interestingly, RT4 cells derived from a BLCA patient with grade I carcinoma [46] express significantly higher levels of both isoforms compared to T24 cells derived from a BLCA patient with grade III carcinoma [47]. These two cell lines exhibit different human leukocyte antigen profiles, growth, migration and metastatic potential, as well as responsiveness to chemotherapeutic drugs [48]. In contrast with what was observed during the analysis of whole BLCA biopsies, in the case of epithelial BLCA cells, the resulting data suggest that more adverse cancerous phenotypes are associated with lower *IL-37c* and *e* mRNA levels. Our combined results may indicate the possible double-faced role of IL-37 and its signaling partners, either protective or pro-tumorigenic depending on its cellular source and target.

UALCAN analysis also supported this, given that *IL-37* levels were 2.89-fold greater in N3 nodal metastasis tumors compared to tumors with fewer than three nearby lymph nodes affected. As mentioned before, also in this case, one needs to take into consideration the relatively small number of patients included in the N3 cohort when extrapolating the data. Furthermore, tumors bearing *TP53* mutation exhibited 1.68-fold increased *IL-37* levels compared to *TP53* non-mutated tumors. *TP53* is a tumor suppressor protein associated with the development of human malignancies [49]. It is the most frequent mutation in BLCA; about half of MIBC tumors bear *TP53* mutations, and in 76% of the samples, the protein is non-functional [50,51]. Indeed, genetic changes in *TP53* are associated with advanced cases and influence disease prognosis and response to therapy in BLCA [52,53,54,55,56]. Regarding *ARID1A*, another BLCA-associated gene [57], no changes in the levels of *IL-37* were detected in mutated vs. non-mutated individuals. The gene codes for a protein member of the SWI/SNF (SWItch/Sucrose Non-Fermentable) chromatin modifying complex, vital for ATP-dependent chromatic remodeling linked to enhanced transcriptional activity, which controls immune recognition as well as other molecular and cellular processes [58,59]. *ARID1A*-inactivating mutations are observed in 20% of BLCA tumors, affecting essential signaling pathways such as PI3K/AKT signaling, which renders the protein a potential target in synergistic therapeutic approaches for the management of BLCA patients [57].

Dramatic *IL-37* changes in BLCA tumors were not accompanied by major alterations in the expression levels of its receptor, *SIGIRR*; only marginal upregulations (~1.3-fold) of its mRNA in cancerous over non-cancerous bladder tissues were observed. Furthermore, papillary tumors exhibited slightly higher levels of the receptor compared to non-papillary ones. Interestingly, *SIGIRR* expression follows the same trend as *IL-37* across the grading of BLCA tumors. Early-stage specimens are characterized by elevated *SIGIRR* levels compared to advanced ones. However, no differences were observed in tumors of different nodal metastases or mutation statuses. Importantly, apart from correlations with pathological parameters of the tumors, our analyses also revealed the prognostic potential of both *IL-37* and *SIGIRR*. High *IL-37* or *SIGIRR* levels have been found to be favorable prognostic factors for the OS of BLCA patients.

Although the exact cellular sources and targets of IL-37 have not been identified in BLCA TME, we attempted to explore whether genetic and/or expression alterations of *IL-37* and *SIGIRR* affect the involvement of various cell types infiltrating the tumor. The use of the TIMER2.0 tool revealed that tumors with non-synonymous, somatic mutations of *IL-37* are characterized by slightly higher infiltration rates of B cells. What is more interesting, however, is the impact of *SIGIRR* mutations on the formation of the TME of BLCA tumors. The most crucial aspect is the dramatic increase in the infiltration by Tregs, possibly indicating the importance of IL-37/SIGIRR signaling in the regulation of immune control in the local microenvironment. It has been stated that BLCA tumors are highly infiltrated by Tregs, correlating with unfavorable prognosis and reduced relapse-free survival (RFS) and response to BCG therapy [60,61]. Additionally, a high CD8^+^ T cell:Tregs ratio indicates better survival and response to chemotherapy [62,63]. Interestingly, Chevalier et al. stated that T24 human bladder cancer cells can enhance the incidence of Tregs and that intratumoral PD-L1^+^ Tregs are specifically induced in patients with NMIBC undergoing BCG, which limits the efficacy of the latter [64]. Other cell subsets affected by non-synonymous, somatic mutations of *SIGIRR* include the marginally upregulated incidence of B cells and DCs and the slightly downregulated incidence of total CD4^+^ T cells and endothelial cells. At this point, it is essential to comment that while the upregulation of Tregs’ distribution is statistically significant, the assumptions drawn need to be carefully evaluated because of the extremely low number of samples in the *SIGIRR*-mutated arm (*n* = 3). Moreover, even though these mutations are anticipated to alter protein function, their exact impact is not known. These associations are based on in silico data, and conclusions about the pathophysiological networks among the aforementioned components, if true, can be made only after pinpointing the exact cell-specific distribution of IL-37 and SIGIRR and the activation status of the involved immunocytes through wet lab experiments.

Complementary to the above approach, we proceeded also to the exploration of possible correlations of *IL-37* expression levels with the infiltration rates of certain immune cell subsets in BLCA tumors. All the linear associations that emerged, although statistically significant, were very minimal; thus, it would be quite risky to develop assessments based on them. However, it is worth commenting that Tregs, neutrophils and CD8^+^ T cells, all critical for the balance between immune surveillance and immune escape, were affected.

Altogether, our data provide evidence for the prognostic and diagnostic potential of *IL-37* and *SIGIRR* in BLCA patients and share indications for their possible involvement in BLCA tumorigenesis and progression. Certain expression patterns associated with lower OS rates, advanced tumor stages and the presence of *TP53* mutations, as described above, lead us to recommend that low *IL-37* expression is associated with more advanced BLCA cases of poorer prognosis. Furthermore, the possible dysfunction of mutated *SIGIRR* is linked to increased recruitment of tumor-suppressive Tregs, which, as described previously, are further linked with unfavorable prognosis and response to treatment [65]. Validation qPCR assays shed light on the specific transcript variant that is deregulated in BLCA: *IL-37e* is significantly higher in tumors of higher grades compared to lower ones, most probably produced predominantly by cells of the TME. On the other hand, human bladder cancer epithelial cells express *IL-37c* and *IL-37e*, which, in this case, are higher in less adverse cellular phenotypes. Interestingly, the confirmation of the expression of the immunoregulatory IL-37 cytokine and its receptor SIGIRR by immune and non-immune cells offers the basis for their future investigation as further possible sites of immune control, like the already-described immune checkpoints [65,66]. Indeed, our previous work revealed that IL-37/SIGIRR signaling shares certain protein members with the PD-1/PD-L1 and CTLA-4 ICs signaling pathways [25]. This may be of utmost importance for the future development of improved treatment approaches in the case of BLCA, which, as a highly immunogenic malignancy, records good clinical responses after IC therapy [46,66].

Nevertheless, this study also has certain limitations including the relatively small number of samples in the validation cohort. It is vital that the resulting data are further confirmed in large clinical groups for the detailed assessment of possible associations with a series of clinicopathological, laboratory and treatment parameters. As previously shown in BLCA patients treated with neoadjuvant chemotherapy followed by radical cystectomy, the integration of cellular, molecular and biochemical data may reveal prognostic and/or predictive factors [67]. Moreover, sequential BLCA biopsies should be evaluated for the possible differential expression signature of the cytokine throughout the course of the disease and/or in conjunction with therapeutic strategies to be recorded. Parallel evaluation of paired peripheral blood and tissue biopsy specimens should be performed to comparatively assess target tissue *c* peripheral blood-specific distributions of IL-37 and SIGIRR (both at the mRNA and protein level), which, like other immunoregulatory factors in inflammatory conditions, may be differentially distributed as a result of the local or systemic control of immune responses [68,69]. It is essential to identify the cellular source(s) and targets of IL-37 (SIGIRR-expressing cells) in order to comprehend the specific effect of any possible aberrations on the intercellular interplays within the BLCA microenvironment, as well as on the prognosis of the course of the disease. Lastly, one always needs to bear in mind that even though IL-37 and its partners may play a protective role against tumor development, such as the possible role indicated in the case of BLCA, there are other types of malignancies where this cytokine acts in favor of tumor growth, mediating a series of processes, and/or is associated with unfavorable predictors of the diseases, as described in the introduction of this manuscript. It also remains to be studied whether different statuses of the same cancer type are accompanied by opposite effects of IL-37 signaling, rendering it either a friend or foe of the tumor.

Regardless of its limitations, the current study supports the critical involvement of IL-37 in BLCA pathogenesis and development. Additionally, it is important to share that to the best of our knowledge, this is the first time that a specific isoform of IL-37, namely *IL-37e*, has been proposed as a potential biomarker for a human malignancy. Moreover, our data confirm that it is essential to fully elucidate the specific sources, targets and the exact role of this cytokine in BLCA immune responses through mechanistic studies at the molecular and cellular levels. Preclinical and clinical studies in well-defined cohorts are needed to exploit IL-37’s and SIGIRR’s potential to serve as novel molecular markers that could facilitate more personalized and precise approaches and ameliorate the prognosis and risk stratification of BLCA patients.

## 4. Materials and Methods

### 4.1. Study Design

To investigate the possible differential distribution of IL-37 and SIGIRR expression in BLCA versus non-BLCA specimens, the UALCAN portal (http://ualcan.path.uab.edu/; data accessed on 1 December 2022) [35] and TNMplot web tool (www.tnmplot.com; data accessed on 20 December 2022) [36] were used. Comparative analyses of IL-37 and SIGIRR expression profiles were performed in (a) BLCA versus non-cancerous bladder biopsies and (b) paired biopsies of BLCA versus adjacent normal tissues of the same patient. The Kaplan–Meier plotter tool (www.kmplot.com; data accessed on 1 December 2022) [37] was utilized to assess the effect of the above mRNA levels on the survival rates of BLCA patients. The differential mRNA distribution between papillary and non-papillary tumors or tumors of various stages, nodal metastases or mutation statuses was explored using the UALCAN portal [35]. The effect of IL-37 or SIGIRR nucleotide changes or expression level aberrations on the differential distribution of various immune cell subsets infiltrating the BLCA tumor was analyzed using the TIMER2.0 webserver (http://timer.cistrome.org/; accessed on 20 December 2022) [38]. Validation of the expression levels of IL-37 was performed in human BLCA cell lines and biopsies applying qPCR assays specifically designed for the assessment of each of the five isoforms. Correlations with the stage of the disease were assessed though appropriate statistical tests.

### 4.2. Assessment of IL-37 and SIGIRR Levels in BLCA versus Non-BLCA Lung Tissues

The TNMplot tool (www.tnmplot.com; accessed on 20 December 2022) [36] was used for the investigation of possible differential expression patterns of IL-37 and SIGIRR in BLCA tumors. RNA sequencing data deposited in The Cancer Genome Atlas (TCGA) database (https://www.cancer.gov/tcga; accessed on 29 December 2022) were processed for the comparative analysis of (a) 411 BLCA versus 30 non-BLCA individuals and (b) 19 pairs of BLCA versus adjacent normal tissue biopsies. The median and range of expression levels of each group, fold changes of the median between groups and the *p*-values of the non-parametric Mann–Whitney U test applied are reported; significant changes were considered those with *p*-values < 0.05 and fold changes >2 or <0.5.

### 4.3. Analysis of Associations between IL-37 or SIGIRR Levels and Certain Pathological Characteristics of the BLCA Tumor

Possible correlations of *IL-37* and *SIGIRR* mRNA levels with BLCA histological type, stage, nodal metastasis, and *TP53* and *ARID1A* mutation status were explored by processing data through the UALCAN portal (http://ualcan.path.uab.edu/; accessed on 20 December 2022) [35]. Significant differences between groups were considered those with fold changes >2 or <0.5 and *p*-values < 0.05.

### 4.4. Exploration of Correlations between IL-37 and SIGIRR Gene Alterations or Expression Levels and Immune Cell Infiltration Patterns in BLCA Tumors

Associations of *IL-37* or *SIGIRR* nucleotide changes and expression levels with the immune cell infiltration of bladder tumors were evaluated through the TIMER2.0 webserver (http://timer.cistrome.org/; accessed on 20 December 2022) [38]. Specifically, the “mutation” module was utilized for the investigation of distribution of macrophages, CD4^+^ T cells, regulatory T cells (Tregs), dendritic cells (DCs), B cells, monocytes and endothelial cells in BLCA tumors of patients with *IL-37* and *SIGIRR* somatic mutations versus those without. Wilcoxon *p*-value and log_2_fold change of infiltration levels between the groups were estimated. The “gene” module was utilized for the exploration of possible correlations between the expression levels of *IL-37* and infiltration rates by CD4^+^ or CD8^+^ T cells, Tregs, γδ T cells, B cells, neutrophils, monocytes, macrophages, DCs, NK, mast cells, cancer-associated fibroblasts, lymphoid, myeloid or granulocyte-lymphocyte progenitor cells, endothelial cells, eosinophils, hematopoietic stem cells and myeloid-derived suppressor cells (MDSCs). Spearman’s rho and *p*-values were calculated for the evaluation of linear correlations.

### 4.5. Human Bladder Cancer Biopsies

A total of 67 bladder tumor samples were analyzed. Specimens were obtained from BLCA patients with primary urothelial carcinoma who underwent transurethral resection (TURBT) of bladder tumors for primary non-muscle-invasive bladder cancer (NMIBC) or radical cystectomy (RC) for muscle-invasive bladder cancer (MIBC) at the “Laiko” General Hospital, Athens, Greece. Upon sampling, tissue was sectioned in mirror images; one of the sections was processed for the histological verification of the presence of urothelial cancer, while the other was immediately frozen in liquid nitrogen and stored at −80 °C until further processing.

Patients received adjuvant therapy according to European Association of Urology (EAU) guidelines [44], while none of them received any form of neoadjuvant treatment prior to surgery. The EORTC calculator was utilized for the risk stratification of the NMIBC patients (www.eortc.be/tools/bladdercalculator/; data accessed on 15 December 2022). The study was performed according to the ethical standards of the 1975 Declaration of Helsinki, as revised in 2008, and approved by the ethics committee of our institution. Informed consent was obtained from all participating patients.

### 4.6. Cell Cultures

Human T24 (transitional cell carcinoma, grade III) [46] and RT4 (transitional cell papilloma, grade I) [47] urinary bladder cell lines were maintained in McCoy’s 5a (modified) medium supplemented with 10% fetal bovine serum (FBS) and 1% penicillin/streptomycin (all from Thermo Fischer Scientific Inc, Waltham, MA, USA), and incubated in a CO_2_-incubator at 37 °C.

### 4.7. Extraction of Total RNA and Reverse Transcription

Total RNA was extracted using TRI-Reagent (Molecular Research Center, Inc., Cincinnati, OH, USA) according to the manufacturer’s instructions. The concentration and purity of the extracted RNA were evaluated spectrophotometrically, while the integrity of the extracted RNA was assessed using agarose gel electrophoresis.

Next, 1.0 μg of total RNA was reverse-transcribed in a 20 μL reaction containing 5 μM oligo-dT primer, 50 U MMLV reverse transcriptase (Invitrogen, Carlsbad, CA, USA), 40 U recombinant ribonuclease inhibitor (Invitrogen) and 0.5 mM dNTPs mix (Invitrogen) at 37 °C for 60 min. Enzyme heat inactivation was performed at 70 °C for 15 min.

### 4.8. Quantitative Real Time PCR (qPCR)

The expression levels of *IL-37* isoforms (*a–e*) were estimated by applying specifically designed SYBRGreen fluorescent-based qPCR assays. Isoform-specific *IL-37* primers were as follows: for *IL-37a*, forward 5′-GGGAAACAGAAACCAAAGGA-3′ and reverse 5′-CCCAGAGTCCAGGACCAGTA-3′; for *IL-37b*, forward 5′-AGCCTCCCCACCATGAATTT-3′ and reverse 5′-ATTCCCAGAGTCCAGGACCA-3′; for *IL-37c*, forward 5′-AGTGCTGCTTAGAAGACCCG-3′ and reverse 5′-CCCTTTAGAGACCCCCAGGA-3′; for *IL-37d*, forward 5′-TGCTGCTTAGAAGGTCCAAA-3′ and reverse 5′-GCTATGAGATTCCCAGAGTCCA-3′; for *IL-37e*, forward 5′- TTGTGGGGGAGAACTCAGGA-3′ and reverse 5′-AAGATCTCTTCTAAGCAGCAC TGG-3′, and the housekeeping gene’s *HPRT1* (hypoxanthine-guanine phosphoribosyltransferase 1) primers were as follows: forward 5′-TGGAAAGGGTGTTTATTCCTCAT-3′ and reverse 5′-ATGTAATCCAGCAGGTCAGCAA-3′. All primers were designed according to their published sequences (NCBI Reference Sequence: NM_173205.2, NM_014439.4, NM_173204.2, NM_173202.2, NM_173203.2 and NM_000194.2, respectively) and in silico specificity analysis, amplifying an 128 bp *IL-37a*-specific, an 114 bp *IL-37b*-specific, an 161 bp *IL-37c*-specific, an 104 bp *IL-37d*-specific, an 82 bp *IL-37e*-specific and an 151 bp *HPRT1*-specific amplicon. The qPCR reactions were performed in the QuantStudio 5 Real-Time PCR System (Applied Biosystems, Carlsbad, CA, USA). The 10 μL reaction mixture included Kapa SYBR Fast Universal 2× qPCR Master Mix (Kapa Biosystems, Inc., Woburn, MA, USA), 100 nM of each PCR primer and 10 ng of cDNA. The thermal protocol consisted of a 3 min polymerase activation step at 95 °C, followed by 40 cycles of denaturation at 95 °C for 3 s and the primer annealing and extension step at 60 °C for 30 s. Melting curve analysis and agarose gel electrophoresis were performed following the amplification to distinguish the accumulation of the specific reaction products from non-specific ones or primer dimers. The 2^−ΔΔCT^ relative quantification (RQ) method was performed for the quantification of mRNA expression levels of the tested samples [70]. *HPRT1* was used as an endogenous reference gene for normalization purposes. Duplicate reactions were performed for each sample, and the average CT value was calculated for the quantification analysis.

### 4.9. Statistical Analysis

Part of the statistical analysis of the data had been performed by the aforementioned web portals. Additional statistical tests processing either data from the above omics repositories or the qPCR wet lab experiment performed in this study included the non-parametric Mann–Whitney U and unpaired *t*-test with *t*-test Welch’s correction for between-two-groups analysis, as well as the ordinary one-way ANOVA test for differences between means. GraphPad Prism 8.4.2 software was used. *p*-values < 0.05 were considered significant.

## Figures and Tables

**Figure 1 ijms-24-09258-f001:**
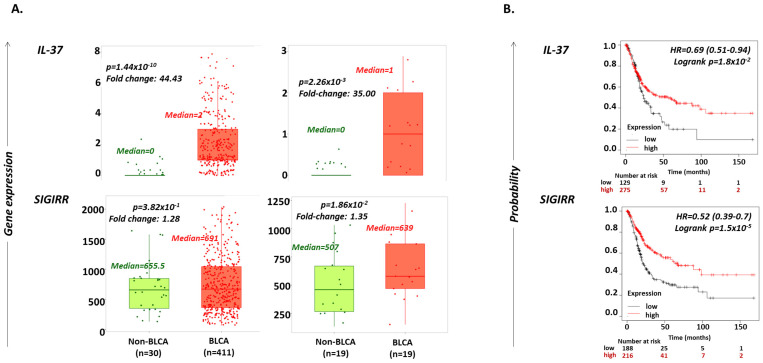
(**A**) Violin plots depicting the differential expression levels of *IL-37* and *SIGIRR* in bladder tissues from BLCA (orange; *n* = 411) vs. non-BLCA (light green; *n* = 30) individuals or paired tumor versus adjacent normal tissues from BLCA patients (*n* = 19 pairs). Median expression levels, Mann–Whitney *p*-values and fold changes between medians are reported. Data were obtained from www.tnmplot.com [36] (accessed on 1 October 2022). (**B**) Kaplan–Meier plots depicting the probability of OS in months in BLCA patients exhibiting high (red) or low (black) expression levels of IL-37 and SIGIRR. Hazard ratio (HR), logrank *p*-values and number of patients with either high or low gene expression, categorized also into those who survived for 50, 100 and 150 months, are reported. Graphs were exported from www.kmplot.com [37] (accessed on 1 December 2022).

**Figure 2 ijms-24-09258-f002:**
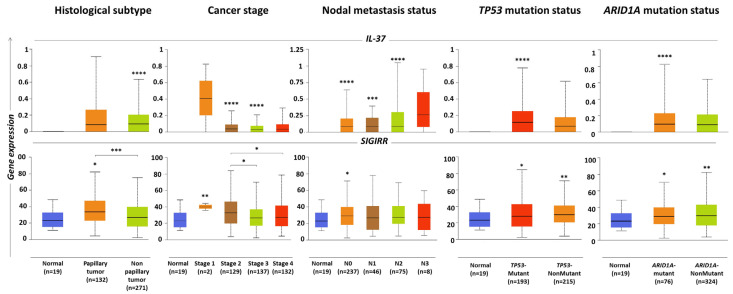
Box and Tukey whisker diagrams showing the differential distribution of *IL-37* and *SIGIRR* expression levels (FPKMs) as analyzed through RNA sequencing in BLCA samples of different histological types, stages, nodal metastases and mutation statuses. Data were obtained from http://ualcan.path.uab.edu/ [38] (accessed on 20 December 2022). Asterisks designate statistically significant differences compared to normal samples, between groups (where accompanied by brackets) as analyzed using the unpaired *t*-test with Welch’s correction or statistically significant linear trends between group means and left-to-right order (where accompanied by an arrow) as analyzed using the one-way ANOVA test; *: *p* < 0.05, **: *p* < 0.01, ***: *p* < 0.001, ****: *p* < 0.0001.

**Figure 3 ijms-24-09258-f003:**
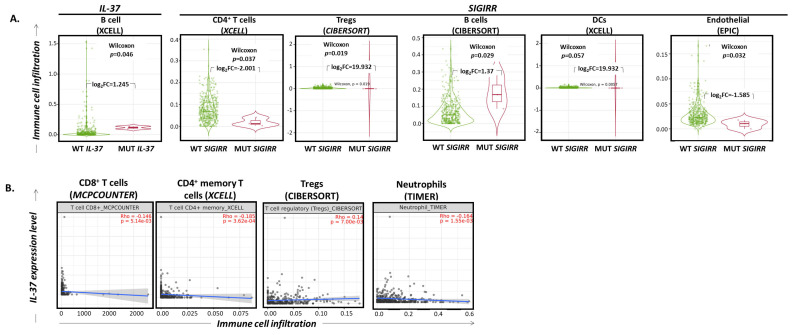
(**A**) Violin plots depicting the distribution of infiltrating B cells in BLCA tumors without versus with mutation on *IL-37* and the distribution of infiltrating CD4^+^ T cells, regulatory T cells (Tregs), B cells, dendritic cells (DCs) and endothelial cells in BLCA tumors without versus with mutation on SIGIRR. Wilcoxon *p*-values and log_2_ (fold changes, FC) are reported. (**Β**) Scatter plot diagrams depicting the linear association between levels of *IL-37* gene expression (log_2_TPM; y-axis) and infiltration of BLCA tumors by CD8^+^ T cells, CD4^+^ T cells, regulatory T cells (Tregs) and neutrophils (x-axis). Spearman’s rho and *p*-values are reported. Data were filtered for tumor purity. Graphs were exported from http://timer.cistrome.org/ [38] (accessed on 20 December 2022).

**Figure 4 ijms-24-09258-f004:**
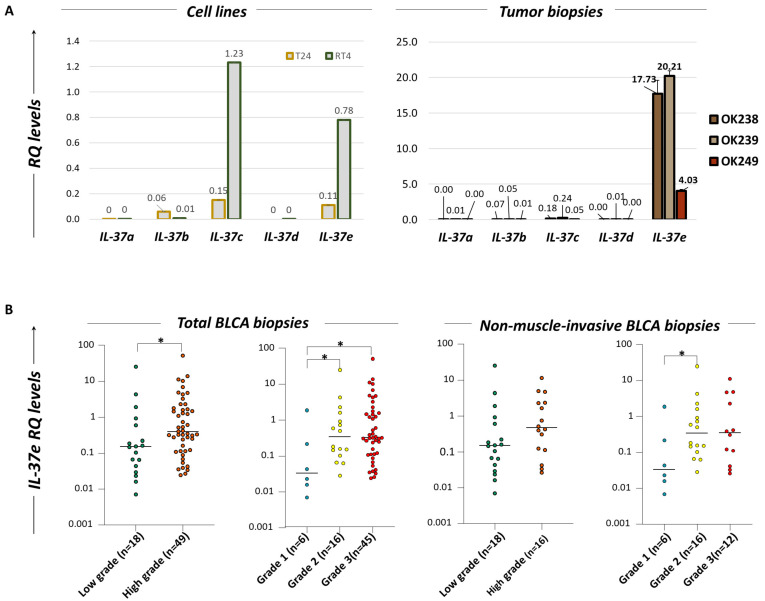
(**A**) Bar diagrams depicting the relative quantification (RQ) mRNA levels of *IL-37 (a-e)* isoforms in human T24 and RT4 BLCA cell lines (left) and indicative BLCA tumor samples (OK238, OK239, OK249) (right), using specifically developed qPCR assays. Data are represented as mean ± standard error (SE) of three independent experiments. (**B**) Dot plot diagrams portraying the differential distribution of *IL-37e* in BLCA tumor biopsies of different grades. Analysis was performed by processing total BLCA samples and non-muscle-invasive samples separately, applying Mann–Whitney *t*-test; *: *p* < 0.05.

**Table 1 ijms-24-09258-t001:** Differential distribution of *IL-37* and *SIGIRR* mRNA levels in BLCA vs. non-BLCA tissues or among BLCA tumors of different histological types, stages, normal metastases or mutation statuses. Number of patients in each group, median and range of expression levels (where available), fold change over non-BLCA or normal samples and *p*-values are reported. Non-BLCA (normal) versus BLCA and adjacent normal vs. tumor comparisons and related statistics were assessed through the Tempol tool (www.tnmplot.com; accessed on 20 December 2022) [36], while normal versus primary tumor as well as comparisons among tumors of different histological subtypes, cancer stages, metastases or mutation statuses were assessed through the UALCAN platform (http://ualcan.path.uab.edu/) [35] (data were accessed 10 December 2022).

	*IL-37*				*SIGIRR*			
Non-BLCA vs. BLCA								
	*n*	Levels (Median; Range)	Fold Change	*p*-Value	*n*	Levels (Median; Range)	Fold Change	*p*-Value
Non-BLCA (normal)	30	0			30	655.5		
BLCA	411	2	2	1.44 × 10^−10^	411	691	1.05	3.82 × 10^−1^
Normal	19	0 (0–0)			19	23.11 (11.29–48.4)		
Primary tumor	408	0.09 (0–0.73)	0.09	NS	408	29.31 (2.30–80.10)	1.27	9.46 × 10^−3^
Tumors vs. adjacent normal								
Adjacent normal	19	0			19	507		
Tumor	19	1	1	2.26 × 10^−3^	19	639	1.26	1.86 × 10^−2^
BLCA histological subtype								
Normal	19	0 (0–0)			19	23.11 (11.29–48.4)		
Papillary	132	0.08 (0–0.91)	0.08	NS	132	33.48 (4.25–82.13)	1.45	2.88 × 10^−2^
Non-papillary	271	0.09 (0–0.63)	0.09	2.06 × 10^−10^	271	26.84 (2.30–75.22)	1.16	NS
Cancer stage								
Normal	19	0 (0–0)			19	23.11 (11.29–48.4)		
Stage 1	2	1.03 (0–2.06)	1.03	NS	2	40.08 (35.86–44.30)	1.73	NS
Stage 2	129	0.1 (0–0.64)	0.1	6.38 × 10^−5^	129	33.13 (3.81–84.25)	1.43	9.77 × 10^−4^
Stage 3	137	0.08 (0–0.52)	0.08	4.02 × 10^−6^	137	26.84 (2.30–70.21)	1.16	NS
Stage 4	132	0.09 (0–0.73)	0.09	NS	132	27.23 (4.33–78.70)	1.18	NS
Nodal metastasis status								
Normal	19	0 (0–0)			19	23.11 (11.29–48.4)		
N0	237	0.09 (0–0.64)	0.09	1.83 × 10^−8^	237	29.31 (2.30–71.13)	1.27	1.45 × 10^−2^
N1	46	0.09 (0–0.4)	0.09	1.82 × 10^−4^	46	26.85 (4.33–77.7)	1.16	NS
N2	75	0.09 (0–1.05)	0.09	1.62 × 10^−7^	75	27.16 (4.53–69.18)	1.18	NS
N3	8	0.26 (0–0.95)	0.26	NS	8	27.39 (5.29–59.47)	1.19	NS
*TP53* mutation status								
Normal	19	0 (0–0)			19	23.11 (11.29–48.4)		
Mutant	193	0.116 (0–0.779)	0.116	5.64 × 10^−14^	193	27.84 (2.3–84.45)	1.20	2.45 × 10^−2^
Non-mutant	215	0.069 (0–0616)	0.069	NS	215	29.68 (3.67–71.13)	1.28	6.38 × 10^−3^
*ARID1A* mutation status								
Normal	19	0 (0–0)			19	23.11 (11.29–48.4)		
Mutant	76	0.096 (0–0.824)	0.096	3.63 × 10^−5^	76	28.93 (2.3–70.52)	1.25	1.80 × 10^−2^
Non-mutant	324	0.091 (0–0.641)	0.091	NS	324	29.69 (3.81–82.13)	1.28	8.96 × 10^−3^

NS: non-significant, N0-4 (www.cancer.gov; accessed on 29 December 2022): the number of lymph nodes that contain cancer.

**Table 2 ijms-24-09258-t002:** Relative quantification (RQ) levels of IL-37e in human BLCA tumor biopsies of various grades, as assessed with specifically designed qPCR assays. Number of samples (N) in each group, median (range) values, fold changes between two groups and Mann–Whitney *p*-values are reported.

	N	Median (Range)		Fold Change	*p*-Value
Total BLCA biopsies					
Low-grade	18	0.14 (0.007–25.18)	High- over low-grade:	2.79	0.0269
High-grade	49	0.39 (0.024–50.83)			
Grade 1	6	0.03 (0.007–1.88)	Grade 2 over Grade 1	11.67	0.049
Grade 2	16	0.35 (0.029–25.18)	Grade 3 over Grade 2	0.94	NS
Grade 3	45	0.33 (0.024–50.85)	Grade 3 over Grade 1	11.00	0.017
Non-muscle-invasive BLCA biopsies					
Low-grade	18	0.14 (0.007–25.18)	High- over low-grade:	2.79	0.0269
High-grade	16	0.39 (0.024–50.83)			
Grade 1	6	0.03 (0.007–1.88)	Grade 2 over Grade 1	11.67	0.049
Grade 2	16	0.34 (0.029–25.18)	Grade 3 over Grade 2	0.94	NS
Grade 3	12	0.35 (0.026–11.34)	Grade 3 over Grade 1	11.00	NS

## Data Availability

The data that support the findings of this study are available from the corresponding author upon reasonable request.

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
