# Peer review of "Deregulated Expression of IL-37 in Patients with Bladder Urothelial Cancer: The Diagnostic Potential of the IL-37e Isoform"

_ijms, 2023, doi:10.3390/ijms24119258_

Round 1
Reviewer 1 Report
The topic of the article is relevant from a practical and theoretical point of view. The research results obtained by the authors of the article and their analysis may be interesting and useful to IJMS readers. However, I recommend that the authors make certain changes to their article.
1. (73-75) “while PTEN (phosphatase and tensin homolog) phosphatase, PI3K (Phosphoinositide 3-kinases) kinase, MAPK (mitogen-activated protein kinase) and other antiinflammatory ones are enhanced [5].” Since when have PI3K and MAPK become anti-inflammatory kinases? Literally in the link [5] it says: “Proteomic and transcriptomic investigations revealed that IL-37 used IL-1R8 to harness the anti-inflammatory properties of the signaling molecules Mer, PTEN, STAT3 and p62(dok) and to inhibit the kinases Fyn and TAK1 and the transcription factor NF-κB, as well as mitogen-activated protein kinases. Furthermore, IL-37-IL-1R8 exerted a pseudo-starvational effect on the metabolic checkpoint kinase mTOR.”. As is currently known, IL-37 is an obvious inhibitor of the PI3K/AKT/mTOR proinflammatory pathway associated with growth and cell proliferation, as well as an inhibitor of the even more proinflammatory MAPK/NF-kB pathway. Please make corrections to the text. It can also be noted that at the same time extracellular IL-37 is an activator of AMP-dependent protein kinase and an activator of the Mer-PTEN-DOK pathways. In addition, IL-37/IL-18Ra/IL1-R8 delays MyD88 and thereby restricts the transmission of signals downstream from IL-1 and TLR.
2. (96-103) “Also, HCC-secreted IL-37 acts on DCs and increases the expression of the surface molecules MHC-II, CD86, and CD40, as well as the secretion of the cytokines IL-2, IL-12, IL-12p70, interferon-a (IFN-α), and IFN-γ, that are further linked to increased recruitment of CD11c+ DCs in the tumor infiltration and suppression of the growth of the tumors [19].” The authors of the article mislead readers by giving an incorrect interpretation of the cited source - [19]. So, in [19] (Chapter: IL-37 indirectly enhances the ability of DCs to induce anti-tumor immune response) it says literally the following: “The proportion of CD86, CD40 and MHC class II on the surface of DCs treated with the supernatant of Hep3B/LV-IL37 cells was significantly higher than on the surface of DCs treated with the supernatant of Hep3B/LV-NC cells, DMEM (nutrient medium) with recombinant IL-37, or DMEM alone. Interestingly, there was no significant difference in the proportion of CD86 and MHC class II on the surface of DCs treated with the supernatant of Hep3B/LV-NC, DMEM with recombinant IL-37, or DMEM alone. In addition, the proportion of CD40 on the surface of DCs treated with the supernatant of Hep3B/LV-NC or DMEM with recombinant IL-37 were decreased”. Thus, we are not talking about the direct effect of IL-37 on DC, but about the fact that recombinant IL-37 cancels the immunosuppressive effect of tumor cells on DC. At the same time, IL-37, when directly acting on DC, lowers CD40 expression in them. In addition, there is a lot of data in the scientific literature on the direct suppressor effects of IL-37 on DC, CTL, Th1 and Th17, M1 macrophages, what can lower antitumor immunity in some cases. This is one of the reasons for the contradictory, ambiguous effect of IL-37 on tumor growth.
3. (245-247). It is not clear how these IL-37 and SIGUR mutations affect the function of these molecules, for example, according to in silico.
4. (267-274) This correlation should be defined as "very weak" (Spearman's rho = 0.00-0.19).
5. According to the latter classification, SIGUR is better designated as IL-1R8.
6. (399-402) It is unclear which cell expression of IL-1R8 is a favorable factor.
7. (408-417) IL-37 is known to stimulate the formation of T-reg [Osborne DG, Domenico J, Fujita M. Expression of IL-37 Induces a Regulatory T-Cell-like Phenotype and Function in Jurkat Cells. Cells. 2022;11(16):2565. doi: 10.3390/cells11162565. PMID: 36010641; PMCID: PMC9406943]. Then how can these data be combined with your ideas about the protective role of IL-37 in BLCA?
8. The interpretation of the results of the study looks one-sided, while the authors, I believe, underestimate the possible negative effects of IL-37 for patients with tumor growth.
Author Response
Response to Reviewer’s 1 Comments
The topic of the article is relevant from a practical and theoretical point of view. The research results obtained by the authors of the article and their analysis may be interesting and useful to IJMS readers. However, I recommend that the authors make certain changes to their article:
Point 1: (73-75) “while PTEN (phosphatase and tensin homolog) phosphatase, PI3K (Phosphoinositide 3-kinases) kinase, MAPK (mitogen-activated protein kinase) and other antiinflammatory ones are enhanced [5].” Since when have PI3K and MAPK become anti-inflammatory kinases? Literally in the link [5] it says: “Proteomic and transcriptomic investigations revealed that IL-37 used IL-1R8 to harness the anti-inflammatory properties of the signaling molecules Mer, PTEN, STAT3 and p62(dok) and to inhibit the kinases Fyn and TAK1 and the transcription factor NF-κB, as well as mitogen-activated protein kinases. Furthermore, IL-37-IL-1R8 exerted a pseudo-starvational effect on the metabolic checkpoint kinase mTOR.”. As is currently known, IL-37 is an obvious inhibitor of the PI3K/AKT/mTOR proinflammatory pathway associated with growth and cell proliferation, as well as an inhibitor of the even more proinflammatory MAPK/NF-kB pathway. Please make corrections to the text. It can also be noted that at the same time extracellular IL-37 is an activator of AMP-dependent protein kinase and an activator of the Mer-PTEN-DOK pathways. In addition, IL-37/IL-18Ra/IL1-R8 delays MyD88 and thereby restricts the transmission of signals downstream from IL-1 and TLR.
Response 1: We thank the reviewer for point out this. We agree that the information given should be reconstructed and corrected. We did that; please see lines 72-74 of the revised manuscript [“PTEN (phosphatase and tensin homolog) phosphatase, PI3K (Phosphoinositide 3-kinases) kinase, MAPK (mitogen-activated protein kinase) and other pro-inflammatory signaling cascades are repressed [4,5]”]. Also, in lines 77-81 information regarding the activation impact of IL-37 on AMP-dependent protein kinase, Mer-PTEN-DOK pathway as well as MyD88 delay due to IL-37/IL-18Ra/IL-1R8 interaction was added: [“What is more, extracellular IL-37 is an activator of AMP-dependent protein kinase and an activator of the Mer-PTEN-DOK (docking protein) pathways. In addition, IL-37/IL-18Ra/IL1-R8 retards MyD88 (myeloid differentiation primary response 88) and thereby harnesses signals transmission downstream TLRs (Toll-like receptor) [16,17]”].
Point 2: (96-103) “Also, HCC-secreted IL-37 acts on DCs and increases the expression of the surface molecules MHC-II, CD86, and CD40, as well as the secretion of the cytokines IL-2, IL-12, IL-12p70, interferon-a (IFN-α), and IFN-γ, that are further linked to increased recruitment of CD11c+ DCs in the tumor infiltration and suppression of the growth of the tumors [19].” The authors of the article mislead readers by giving an incorrect interpretation of the cited source - [19]. So, in [19] (Chapter: IL-37 indirectly enhances the ability of DCs to induce anti-tumor immune response) it says literally the following: “The proportion of CD86, CD40 and MHC class II on the surface of DCs treated with the supernatant of Hep3B/LV-IL37 cells was significantly higher than on the surface of DCs treated with the supernatant of Hep3B/LV-NC cells, DMEM (nutrient medium) with recombinant IL-37, or DMEM alone. Interestingly, there was no significant difference in the proportion of CD86 and MHC class II on the surface of DCs treated with the supernatant of Hep3B/LV-NC, DMEM with recombinant IL-37, or DMEM alone. In addition, the proportion of CD40 on the surface of DCs treated with the supernatant of Hep3B/LV-NC or DMEM with recombinant IL-37 were decreased”. Thus, we are not talking about the direct effect of IL-37 on DC, but about the fact that recombinant IL-37 cancels the immunosuppressive effect of tumor cells on DC. At the same time, IL-37, when directly acting on DC, lowers CD40 expression in them. In addition, there is a lot of data in the scientific literature on the direct suppressor effects of IL-37 on DC, CTL, Th1 and Th17, M1 macrophages, what can lower antitumor immunity in some cases. This is one of the reasons for the contradictory, ambiguous effect of IL-37 on tumor growth.
Response 2: We deeply thank the reviewer for identifying this error from our side. We amended the corresponding text as follows: “Also, recombinant IL-37 hampers the immunosuppressive effect of HCC cells against DCs, which is demonstrated by the increased expression of the surface molecules MHC-II, CD86, and CD40, while stimulates DCs to secrete IL-2, IL-12, IL-12p70, inter-feron-a (IFN-α), and IFN-γ, further enhancing that anti-tumor effect of T cells. What is more, HCC overexpression of IL-37 is associated with increased recruitment of CD11c+ DCs in the tumor infiltration and suppression of the growth of the tumors [22].” (lines 106-112 in the introduction). Also, we added information about the direct anti-inflammatory effects of IL-37 within the inflammatory network (as mentioned by the reviewer) in lines 81-85 of the revised manuscript.
Point 3: (245-247). It is not clear how these IL-37 and SIGUR mutations affect the function of these molecules, for example, according to in silico.
Response 3: We thank the reviewer for giving the opportunity to make this more clear. Even though, in the first version of the manuscript, we mentioned that these are non-synonymous somatic mutations (lines: 265, 267, 421, 432) (this was the only information provided by the TIMER2.0 tool), after Reviewer’s comment we added the following in the discussion section (lines: 437-442): “Besides, even though these mutations are anticipated to alter protein function, their exact impact is not known. These associations are based on in-silico data and conclusions about the pathophysiological networks among the aforementioned compartments, if true, could be made only after pinpointing the exact cell-specific distribution of IL-37 and SIGIRR and the activation status of the involved immunocytes, through wet-lab experiments.”. By this addition, we aimed at signifying the need for further in-silico investigation of the impact of these mutations in the interaction among immune and tumor cells.
Point 4: (267-274) This correlation should be defined as "very weak" (Spearman's rho = 0.00-0.19).
Response 4: We thank the reviewer for mentioning this. We added the “very weakly” in line 286 of the revised manuscript.
Point 5: According to the latter classification, SIGUR is better designated as IL-1R8.
Response 5: We agree with the reviewer. We have mentioned that in line 67 of the introduction.
Point 6: (399-402) It is unclear which cell expression of IL-1R8 is a favorable factor..
Response 6: Yes, we agree with the reviewer. However, the information provided are based on online repositories that do not clarify that, since they don’t include such data. That is the reason, we discussed the need for clarification of the cell types expressing IL-37 and/or SIGIRR in the discussion section (lines 480-485). In this regard and to be sure that this is clear to the reader, we furthermore added “(SIGIRR-expressing cells)” and “as well as on the prognosis of the course of the disease”, in lines 484 and 486 of the revised manuscript, respectively.
Point 7: (408-417) IL-37 is known to stimulate the formation of T-reg [Osborne DG, Domenico J, Fujita M. Expression of IL-37 Induces a Regulatory T-Cell-like Phenotype and Function in Jurkat Cells. Cells. 2022;11(16):2565. doi: 10.3390/cells11162565. PMID: 36010641; PMCID: PMC9406943]. Then how can these data be combined with your ideas about the protective role of IL-37 in BLCA?.
Response 7: We really appreciate the meticulous reviewer’s work on our manuscript, and we thank him for giving us the opportunity to clarify this. Indeed, there are data supporting the anti-tumoral but also the pro-tumoral effect of IL-37 via mechanisms involving immune cells but also other compartments of the TME. To that end, we replaced the word “protective” with the word “regulatory” in line 86 of the revised manuscript. What is more, we added information from Osborne DG, et al (Expression of IL-37 Induces a Regulatory T-Cell-like Phenotype and Function in Jurkat Cells. Cells. 2022;11(16):2565), as suggested by the reviewer in lines 116-119. The tumor-promoting effect of IL-37 was already mentioned (even though maybe not clear) in lines 112-115, describing its association with the risk of tumor development in CRC experimental mice. We added: “in this case supporting its tumor promoting effect” (lines 115-116), to highlight it. What is more, the correlation of IL-37 expression with poor disease prognosis and/or unfavourable factors in AML, OSCC or breast cancer patients, possibly associated with a pro-tumoral pathogenetic role, is described in lines 141-152.
Point 8: The interpretation of the results of the study looks one-sided, while the authors, I believe, underestimate the possible negative effects of IL-37 for patients with tumor growth.
Response 8: Similarly, to the comment above, we are taking the opportunity reviewer gave to us to clarify that not in all cases IL-37 seems to have a protective effect against tumors. So, apart from the abovementioned additions/amendments in the text, we also added the following concluding remark in lines 486-493 of the revised manuscript: “Lastly, one always needs to bear in mind that even though IL-37 and its partners may possess a protective role against tumor development, like possible indicated in the case of BLCA, there are other types of malignancies where this cytokine acts in favor of tumor growth mediating a series of processes and/or is associated with unfavorable predictors of the diseases, as described in the introduction of this manuscript. It also re-mains to be studied, whether different statuses of the same cancer type, are accompanied by opposite effects of IL-37 signaling rendering it either friend of foe or the tumor.”
Reviewer 2 Report
This article talks about : Cellular and molecular immune compartments play a crucial role in the development and perpetuation of human malignancies shaping anti-tumor responses. A novel immune regulator is interleukin-37 (IL-37) and its isoforme, especially for highly tumorigenic tumors such as bladder urothelial carcinoma (BLCA). interleukin 37 and its receptor SIGIRR have a prognostic and diagnostic value against various tumor pathologies and their expression can increase or decrease depending on the tumor being considered: Bioinformatics analysis revealed that IL-37 levels correlate to BLCA tumor development and are higher in patients with longer overall survival. These data strongly indicate the need for further investigation of the involvement of this cytokine and interconnected molecules in the pathophysiology of the disease and its prospective as a therapeutic target and biomarker for BLCA.
My suggestions:
- English language should be improved in both grammar and syntax
- Lines 140-161: in addition to the expression of interleukin 37 and its receptor SIGIRR, it could be useful to study the expression of other genes both in bladder cancer and in the other tumor pathologies addressed in the article. At this regard I suggest the following article: https://pubmed.ncbi.nlm.nih.gov/36898352/
- The study analyzes the expression of the interleukin 37 gene and its receptor associated with OS and patient mortality. It would be interesting to integrate the genetic analysis with a biochemical analysis. At this regard i suggest this article: https://pubmed.ncbi.nlm.nih.gov/34258022/
English language should be improved in both grammar and syntax
Author Response
Response to Reviewer’s 2 Comments
This article talks about: Cellular and molecular immune compartments play a crucial role in the development and perpetuation of human malignancies shaping anti-tumor responses. A novel immune regulator is interleukin-37 (IL-37) and its isoforme, especially for highly tumorigenic tumors such as bladder urothelial carcinoma (BLCA). interleukin 37 and its receptor SIGIRR have a prognostic and diagnostic value against various tumor pathologies and their expression can increase or decrease depending on the tumor being considered: Bioinformatics analysis revealed that IL-37 levels correlate to BLCA tumor development and are higher in patients with longer overall survival. These data strongly indicate the need for further investigation of the involvement of this cytokine and interconnected molecules in the pathophysiology of the disease and its prospective as a therapeutic target and biomarker for BLCA.
My suggestions:
Point 1: English language should be improved in both grammar and syntax.
Response 1: We thank the reviewer for this comment. The manuscript was checked by an English-native speaker.
Point 2: Lines 140-161: in addition to the expression of interleukin 37 and its receptor SIGIRR, it could be useful to study the expression of other genes both in bladder cancer and in the other tumor pathologies addressed in the article. At this regard I suggest the following article: https://pubmed.ncbi.nlm.nih.gov/36898352/.
Response 2: We thank the reviewer for this comment, and we agree that also other pivotal compartments within the BLCA tumor microenvironment should be further explored in future studies. For that reason, we made a relevant comment and added the suggested reference in lines 163-164 of the introduction: “…similarly to other partners of the BLCA TME, e.g. the fibroblast growth factor receptor 3 (FGFR3) that were very recently suggested as possible therapeutic targets [37].”.
Point 3: The study analyses the expression of the interleukin 37 gene and its receptor associated with OS and patient mortality. It would be interesting to integrate the genetic analysis with a biochemical analysis. At this regard i suggest this article: https://pubmed.ncbi.nlm.nih.gov/34258022/.
Response 3: We agree with the reviewer that future studies should be focused on the integration of genetic, molecular, and biochemical parameters for the development of a panel of survival biomarkers of the disease. To that end, we added: “As previously shown in BLCA patients treated with neoadjuvant chemotherapy followed by radical cystectomy, integration of cellular, molecular and biochemical data may reveal prognostic and/or predictive factors [74].” (lines 473-476 in discussion section).
Round 2
Reviewer 1 Report
I am completely satisfied with the answers of the authors of the article. I believe that the article in the presented form is relevant and will be useful for many readers of the journal. I also want to wish the authors further achievements in the scientific direction indicated in their article.
Reviewer 2 Report
Authors answered all comments and suggestions.
Minor revisions are required.